# Possible Therapeutic Mechanisms and Future Perspectives of Vaginal Microbiota Transplantation

**DOI:** 10.3390/microorganisms11061427

**Published:** 2023-05-29

**Authors:** Maimaiti Tuniyazi, Naisheng Zhang

**Affiliations:** Department of Clinical Veterinary Medicine, College of Veterinary Medicine, Jilin University, Changchun 130062, China; mmttn18@mails.jlu.edu.cn

**Keywords:** vaginal microbiota transplantation, dysbiosis, treatment, mechanism

## Abstract

Microbial communities inhabiting the human body play a crucial role in protecting the host against pathogens and inflammation. Disruptions to the microbial composition can lead to various health issues. Microbial transfer therapy (MTT) has emerged as a potential treatment option to address such issues. Fecal microbiota transplantation (FMT) is the most widely used form of MTT and has been successful in treating several diseases. Another form of MTT is vaginal microbiota transplantation (VMT), which involves transferring vaginal microbiota from a healthy female donor to a diseased patient’s vaginal cavity with the goal of restoring normal vaginal microbial composition. However, VMT has not been extensively studied due to safety concerns and a lack of research. This paper explores the therapeutic mechanisms of VMT and discusses future perspectives. Further research is necessary to advance the clinical applications and techniques of VMT.

## 1. Introduction

The human body is home to diverse microbial communities known as the microbiota, which consist of bacteria, archaea, fungi, viruses, and protists. They residue on surfaces and niches that are directly linked or not linked to the external environment [1]. These communities vary greatly in composition and function among different body sites and individuals [2]. Indeed, Techniques such as 16S rRNA sequencing and statistical methods have revealed that every part of the human body is colonized with unique microbial communities that differ in composition and function based on the anatomical niche and health status of individuals. For example, the oral and nasal cavity [3,4], lung [5], and skin [6], and the gastrointestinal [7], urinary [8,9], and reproductive [10] tracts harbor specific types of microbiota based on their functions and surfaces.

Traditionally, all microbiomes were thought to be pathogens and cause diseases [11], but it is now understood that they play a crucial role in protecting against pathogens and regulating the host’s inflammatory responses [12]. However, changes (dysbiosis) in normal bacterial communities impair the normal function of microbiota as protectors and modulators and lead to disease reactions. Therefore, maintaining a healthy host microbiota community is crucial. This led to the development of microbial transfer therapy (MTT), which involves replacing diseased microbiomes with healthy ones. The most popular MTT is fecal microbiota transplantation (FMT), which has been used since the 4th century and has gained attention after its approval by the US Food and Drug Administration for treating *Clostridium* difficile infection (CDI) in 2012 [13]. FMT has been found to be effective and safe in treating CDI [14], as well as gastrointestinal and psychiatric disorders such as ulcerative colitis and mental illness [15,16].

These successful applications of FMT inspired a new set of MTT—vaginal microbiota transplantation (VMT). VMT is an emerging experimental medical intervention that aims to restore the otherwise imbalanced vaginal microbiota by transferring vaginal microbiota from a healthy donor to the vaginal cavity of a diseased patient to restore its overall diversity, stability, normal composition, and function [17]. Recently, VMT has been successfully applied to treat bacterial vaginosis without any adverse effects [18]. Briefly, in this clinical trial, five patients (aged 27–47 years old) suffering from symptomatic, intractable, and recurrent bacterial vaginosis were treated with VMT. Among them, 80% of the patients were fully recovered during this study (5–21 months after VMT). Successful treatment results show significant symptom improvement, adherence to Amsel criteria, observation of improved vaginal fluid under a microscope, and restoration of a *Lactobacillus*-dominated vaginal microbiome. However, the biggest drawback was that this study was small in sample size and uncontrolled. In order to discuss the feasibility of manipulating vaginal microbiota by VMT, Gardner and Dukes transferred *Gardnerella vaginalis* from the vaginas of infected women into the vaginas of healthy volunteers who successfully developed the disease [19]. Although these two studies have shown the possibility of manipulating vaginal microbiota by means of transplanting the entire vaginal microbiota or bacterial strain, the research and clinical applications of VMT are still in their infancy. 

Studies have shown that female reproductive tract health is not only maintained by the vaginal microbial community [20,21] but also heavily dependent on healthy intestinal flora [22]. Therefore, the potentiality of using FMT as a tool for treating female reproductive tract diseases, which aims to restore gut microbiota, was discussed [23]. However, compared to the gut microbiota, the vaginal microbiota is rarely explored as an MTT, which resulted in underappreciated clinical applications of VMT.

Safety concerns and lack of case-controlled studies, and regulatory approvals are the main limiting factors of the clinical application of VMT. For this regard, in this paper, we aimed to explore the possible mechanisms of therapeutic effect in VMT to encourage its wider clinical applications as well as future perspectives that could be a direction for further studies.

## 2. Normal Composition and Function of Vaginal Microbiota

The vaginal cavity is a crucial part of the female reproductive system, extending from the cervix and uterus to the external genitalia (vulva) (Figure 1). It is a niche in the human body that harbors a unique microbial community, predominantly composed of *Lactobacillus* spp. [24], including species such as *Lactobacillus crispatus*, *Lactobacillus gasseri*, *Lactobacillus iners*, and *Lactobacillus jensenii* [25,26]. However, other species such as *Bacteroides* spp., *Fusobacterium* spp., *Veillonela* spp., *Actinomycetes* spp., *Bifidobacterium* spp., *Peptococcus* spp., *Peptostreptococcus* spp., *Propionibacterium* spp., *Staphylococcus aureus*, *Staphylococcus epidermidis*, *Streptococcus viridans*, *Enterococcus faecalis*, *Gardnerella vaginalis*, and *Prevotella bivia* [25,26,27], also exist at low levels.

Similar to any other microbiota-residue niches (e.g., gut microbiota), the vaginal microbiota may also interact with the host immune system and act as a protector and modulator against pathogenic agents and inflammatory responses in the vaginal cavity [28]. The presence of *Lactobacillus crispatus* and *Lactobacillus jensenii* in the vagina has been linked to lower levels of cellular inflammation markers and higher levels of anti-inflammatory cytokines such as IL-1a and IL-8, according to a study [29]. Another study found that higher levels of secretory leukocyte peptidase inhibitor (SLPI), an antimicrobial peptide that is typically depleted in women with conditions such as bacterial vaginosis [30,31], can be observed in women with high levels of *Lactobacillus iners* [32]. Doerflinger and colleagues` research suggests that while *Lactobacillus iners ATCC 5195* does activate pattern-recognition receptor (PRR) signaling pathways in human primary vaginal epithelial cells, *Lactobacillus crispatus* ATCC 38820 does not significantly upregulate [33]. These findings indicate that the composition of the vaginal microbiome plays a unique role in maintaining vaginal health.

*Lactobacillus species* in the vaginal play a critical role in maintaining female reproductive health through various directive and indirective anti-pathogenic mechanisms. These mechanisms include producing compounds that directly kill or inhibit pathogens, creating a microbial barrier that attaches to the epithelium and prevents pathogenic agents from adhering, and activating the host’s defense mechanisms against pathogens. These functions demonstrate the unique and crucial role that the composition of the vaginal microbiome plays in maintaining a healthy reproductive system.

Furthermore, a study involving a group of asymptomatic young South African women revealed that the composition of vaginal microbiota is closely related to host genital inflammation [34]. This study found that women with high diversity and low abundance of *Lactobacillus* in their vaginal microbiota experienced higher levels of pro-inflammatory cytokines in the genital area. This highlights the crucial role that the composition and diversity of the vaginal microbiome play in maintaining female reproductive health.

*Lactobacillus species*-dominated vaginal microbiota essential to female reproductive health, and its presence may protect against urological diseases such as bacterial vaginosis, yeast infections, STDs, urinary tract infections, and HIV [35,36,37,38,39,40,41,42,43,44,45,46,47]. A healthy and diverse composition of the vaginal microbiota is important for maintaining gynecologic wellness in women.

In a healthy state, the vaginal environment is maintained by a delicate balance between different elements, including lactic acid production. Lactic acid is crucial in maintaining vaginal homeostasis and preventing the growth of pathogens. There are two sources of lactic acid in the vagina, the first being produced by the vaginal epithelium through the production of L-lactate, which accounts for 20% of total lactic acid. The second source is the vaginal microbiota, responsible for metabolizing glycogen and producing the majority of lactic acid, primarily in the form of D-lactic acid, which accounts for 80% of the total lactic acid [48,49] (Figure 2).

## 3. Development of the Vaginal Microbiota

While the diversity of the microbiota in a healthy women’s vaginal cavity is relatively low, its composition undergoes a series of changes throughout the female life cycle, from childhood to the menopause stage (Figure 3). For example, in childhood, the vaginal microbiota is most diverse and comprises gram-negative anaerobic, gram-positive anaerobic, and aerobic bacteria [54,55]. After childhood—in prepuberal, puberty, and adult stages, the vaginal microbiota becomes less diverse and dominated by *Lactobacillus* spp. [55,56]. In the menopause stage, the vaginal microbiota is also dominated by *Lactobacillus* spp. but more diverse compared to the previous three stages [57]. Although the exact purpose and function of such changes are not clear, it is possibly associated with the reproductivity of a female. Therefore, the age and reproduction status of a female should be considered when carrying out clinical trials and studies regarding female healthy genitalia microbiota composition.

## 4. Factors Related to Changes in Vaginal Microbiota

A healthy vaginal microbiota community plays an important role not only in preventing pathogenic agents from invasion but also in maintaining the female reproductive and gynecologic health and overall host well-being. 

Imbalances of vaginal microbiota are associated with several adverse conditions such as preterm birth, pelvic inflammatory disease, increased risk and transmission of sexually transmitted infection, infertility, and multiple stigmatizing symptoms that impact female health [58]. Therefore, in order to further manipulate the vaginal microbiota in a beneficial direction, it is important to understand the factors associated with changes in the vaginal microbiota community. According to previous studies, there are many factors may affect the vaginal microbiota communities, including but not limited to diseases (bacterial vaginosis), age, hormone physiology (newborn, childhood, puberty, reproductive stage, postmenopausal stage), ethnicity, tobacco, stress, sexual activity, lifestyle and daily practices, probiotics, diet, and exercise [21,22,59,60,61,62,63,64,65,66,67,68].

Regardless of the specific reason, a dysbiosis (imbalance) vaginal microbiota is characterized by lowered *Lactobacillus* spp. and increased anaerobic microorganisms in the vaginal cavity. These ultimately resulted in transformations of the vaginal microbiota composition from *Lactobacillus* spp. to potentially pathogenic facultative anaerobic bacteria and increased vaginal pH (>4.5). 

A healthy vaginal microbiota community composed mainly of Lactobacillus species helps maintain female reproductive health by preventing pathogenic agents from invading. However, disruptions to this balance, caused by factors such as antibiotics, hormonal changes, and sexual activity, can lead to the overgrowth of potentially pathogenic microorganisms and increase the risk of conditions such as bacterial vaginosis, aerobic vaginitis, and sexually transmitted infections, such as human immunodeficiency virus (HIV-1), human papillomavirus (HPV) infection [69], and *Chlamydia trachomatis* infection [70].

## 5. Possible Mechanisms of Vaginal Microbiota Transplantation

Here, we proposed three possible mechanisms of action involved in the VMT therapy, including increased competition for nutrition, increased production of bactericide, virucide and hydrogen peroxide (H_2_O_2_), and specific adhesion to epithelial cells. These are based on the role of healthy vaginal microbiota in maintaining vaginal health, as well as are adapted from previous studies that discussed possible mechanisms of FMT in humans and dogs [71,72].

Similar to the mechanism of FMT in CDI treatment, where the introduction of non-toxigenic *Clostridium difficile* strains can lower the recurrence of CDI in subjects [73], competition for nutrition is the first possible mechanism of VMT treatment (Figure 4). Its main idea is that survival is the first priority of any living organism; microbiomes—healthy or pathogenic—are no exception, which requires absorbing nutrition. Under normal circumstances, although pathogenic and opportunistic microorganisms exist in the vaginal cavity, compared to the healthy vaginal microbiota, they are small in numbers. Therefore, the harmful microbes cannot overcome nutritional competition between the healthy vaginal microbiota. As a result, pathogenic and opportunistic microorganisms are not able to overgrow or lead to disease reactions. 

However, when the vaginal microbiota is disrupted, the vagina creates a suitable microenvironment for surviving and proliferation of pathogenic and opportunistic microorganisms. This kind of disruption further leads to a decrease in the relative abundance of healthy vaginal microbiota that loses advantages for nutritional competition. After the disease-leading microorganisms became the predominant species in the vagina, they absorb most of the nutrition, grow and cause disease.

Transferring healthy vaginal microbiota that predominated by *Lactobacillus* spp. from a healthy donor increases the relative abundance of the healthy microbial community in the vaginal cavity. The increased overall number of healthy microbiota after the VMT procedure gives the advantage of competing nutrition between harmful microorganisms. In this situation, the healthy vaginal donor strains may compete for the same nutrition that is available in the vaginal cavity more successfully than the recipient’s pathogenic strains. This leads to a decrease in the relative abundance of pathogenic agents to a level at which they are no longer able to cause disease reactions. This also indicates healthy *Lactobacillus* spp. offered by VMT can prevent harmful microorganisms from absorbing nutrition. However, this is a slow process, and repeated treatments may be needed for successful outcomes. 

Another possible mechanism of VMT is increased bactericidal and virucidal products. As previously stated [74], it is hypothesized that the predominated vaginal microbiota, *Lactobacillus* spp., plays an important protective role in the vagina by producing bactericides and virucides, including lactic acid and bacteriocins, which prevent the overgrowth of pathogens and other opportunistic microorganisms. In addition, a previous study also suggested that hydrogen peroxide (H_2_O_2_) produced by *Lactobacilli* plays a secondary role in the vaginal microbiota [75]. 

Under normal situations, the volume of bactericidal, virucidal, and H_2_O_2_ compounds produced by *Lactobacillus* spp. are enough to inhibit invaded pathogens from proliferation and causing diseases. However, after the vaginal microbiota is disrupted, *Lactobacillus* spp. become less abundant, which consequently results in decreasing production of such disease-inhabiting agents. Decreased volume of these products, the vaginal cavity becomes more susceptible to disease reactions caused by pathogens and opportunistic microorganisms.

After the patient receives vaginal microbiota from a healthy donor, which is rich in *Lactobacillus* spp., the relative abundance of *Lactobacilli* in the vaginal cavity increases, followed by increasing production of bactericides, virucides, and H_2_O_2_, which consequently inhibit and/or slow down colonization and proliferation of harmful microorganisms (Figure 5).

*Lactobacillus* spp. can inhibit the attachment and colonization of pathogenic agents by producing compounds that directly kill or inhibit pathogens, creating a microbial barrier on the epithelium and stimulating host defense mechanisms. Additionally, several in vitro studies have shown that *Lactobacillus* spp. can prevent the attachment of pathogens on the surface of epithelial cells, including *E. coli*, *Gradnerella vaginalis*, *Klebsiella pneumonia*, *Pseudomonas aeruginosa*, *Staphylococcus aureus*, *group B streptococci*, and *Trichomonas vaginalis* [76,77,78,79,80] (Figure 6). 

All of these three mechanisms are closely related to the *Lactobacillus species* community in the vagina and its pH level. The pH is a very important parameter for bacterial survival. Various niches located in different parts of the human body have different pH features, which create ideal living environments for resident bacterial communities. The presence of lactic acid in the vagina helps maintain its acidic environment with a pH level of around 3.5–4.5, which is crucial for the health and balance of the vaginal microbiota.

The vaginal cavity, being directly connected to the external environment, is highly susceptible to the invasion of pathogenic bacteria. However, the presence of *Lactobacillus* spp. that produces lactic acid helps to maintain a low and acidic vaginal pH of around 3.5–4.5, creating a protective environment for the mucosa that limits the growth of pathogenic microorganisms, including uropathogenic *E. coli*, *Neisseria gonorrhoeae*, and *Chlamydia trachomatis* [28,81,82,83,84,85].

A disrupted vaginal microbiota can lead to an increase in vaginal pH, making the environment more susceptible to diseases. For example, Brotman and colleagues have shown that a higher vaginal pH of more than 4.6 is highly associated with an increased risk of trichomonal, gonococcal, and chlamydial infections, as opposed to a lower pH of less than 4.0 [86]. 

In conclusion, a healthy vaginal microenvironment characterized as rich in *Lactobacillus* spp. and a low pH, which collectively create a protective barrier against pathogen invasion, is of utmost importance for maintaining female reproductive health. Therefore, transferring healthy vaginal microbiota from a donor restores the healthy vaginal cavity that is predominated by *Lactobacillus* spp. and lowers the pH that negatively impacts the survival of pathogenic and opportunistic microorganisms and also reconstructs the protective barrier and prevents harmful bacteria from continuing invasion.

However, it should be noted that although we explored the possible mechanisms of VMT treatment individually aimed to better understand, the therapeutic effect of VMT is a result of combined effort that includes increased competition for nutrition, decreased pH level, and increased production of bactericides and virucides at the same time.

## 6. Risks and Limitations

The main risk of VMT is the possible transmission of pathogenic and opportunistic microorganisms. Indeed, considering such risks, safety issues are the main limitations of VMT therapy in clinics. Therefore, careful screening of the donors for VMT is of utmost importance to avoid exposure to infectious agents. For example, as previously mentioned, pathogenic bacteria can be transmitted by VMT procedure and cause disease pathogenesis [19].

Presently, the donor selection process is focused on safety by excluding as many risky elements as possible to obtain relatively ‘healthy’ vaginal microbiota, which is characterized by a high abundance of *Lactobacillus* spp. 

In addition, the healthy vaginal microbiota is also composed of fungi and viruses [87,88,89,90], such as *Candida albicans*, double-stranded DNA viruses, undefined viruses, and a small proportion of single-stranded DNA viruses. Without a doubt, they have an impact on VMT efficacy. However, current research is mainly focused on bacteria, and more studies are needed to explore the roles of fungi and viruses on the efficacy of VMT treatment. At the same time, other bacterial species that are in low abundance may play a certain role in maintaining vaginal health. However, in current studies, they are mostly ignored. In future studies, the role of such microorganisms could be deeper studied, and it is also a way of increasing the safety and efficacy of VMT treatment.

There is no guarantee that VMT can treat all vaginal disorders. For example, present VMT treatment is limited to bacterial vaginosis [18]. It is unknown if VMT is effective for treating viral vaginosis. Therefore, medical specialists should offer a detailed explanation of risks and limitations that may involve VMT treatment before the procedure.

## 7. Future Perspectives

In the future, VMT will be more widely used for treating diseases such as bacterial vaginosis as more studies are being conducted to explore its efficacy and safety. In addition, other issues such as HIV, HPV, and sexually transmitted diseases are reported to be able to alter vaginal microbiota communities [34,35]. Based on these, restoring the vaginal microbiota by means of VMT may prevent such diseases from further damaging reproductive health.

Similar to FMT treatment, selecting the most suitable donor is the most important part of VMT therapy. Except for regular examinations, such as blood examinations and the possibility of various virus infractions, we are now able to choose the donor one step further based on the vaginal microbiota content with the help of 16S rRNA sequencing technology. The most suitable donor, in theory, is not only healthy but also has the most abundant *Lactobacillus* spp., which plays a crucial role in maintaining vaginal health. In this way, the VMT treatment would be safer and more efficient.

In FMT, studies explored the efficacy and safety of readily available and capsulized fecal materials in treating CDI and observed promising results [91]. The most important parameter in storing bacteria is temperature. For example, Burz and colleagues found that fecal microbiota can be stored at 4 °C for 24 h and don’t lose their viability [92]. However, −80 °C is more suitable for long-term (over 3 months) stool storage. The bacterial viability may diminish dramatically, especially in gram-negative bacterial communities, when stored at room temperatures for more than 8 h. Therefore, the storage conditions for preserving the most viable bacteria are an important part of microbiota transplant therapy.

In the future, VMT may become a widely available treatment option similar to FMT, which may be followed by increased demand that requires more easily accessible vaginal microbiota. Therefore, preparing a vaginal microbiota bank similar to stool banks may be an innovative approach for future studies. However, as stated, the optimal temperature that can preserve the most viability of *Lactobacillus* spp. should be prioritized for building vaginal microbiota banks.

In clinics, when treating CDI with FMT, doctors use laxatives to clear the intestine microbiota; this not only increases vision in endoscopic delivery but also increases the colonization of newly transplanted fecal microbiota to the gastrointestinal tract. Other studies also showed that antibiotic administration prior to CDI treatments could significantly increase FMT efficacy. Although it is impossible to use laxatives to get rid of the vaginal microbiota, it may enhance the efficiency of VMT if a vaginal douche is applied before transplanting microbiota. However, studies are needed before such methods are adapted to clinical applications.

At the same time, studies also found that there are distinct microbial communities between healthy and women with reproductive problems. For example, a recent study involving 31 female participants revealed that they have different vaginal microbiota compositions [93]. This study showed that the vaginal microbiota was primarily characterized by the dominance of the genus *Lactobacillus*, specifically, *Lactobacillus iners* AB-1, which was the most prevalent species among all the groups. Compared with the infertile cohort, the healthy group exhibited an excessive growth of anaerobic bacteria, including *Leptotrichia* and *Snethia*, which are associated with vaginal dysbiosis. 

Another recent study also indicated that the composition of vaginal microbiota but not of seminal microbiota is associated with successful intrauterine insemination in couples with idiopathic infertility, where the domination of *Lactobacillus crispatus* was strongly associated with successful pregnancy [94]. Although we cannot overlook the fact that these women may have other issues that induce their infertility, the distinct differences between vaginal microbiota communities may offer theoretical bases for exploring whether transferring vaginal microbiota from a healthy fertile donor leads to a reconstruction of vaginal microbiota in an infertile woman and further leads to increased pregnancy rate. However, again, such a hypothesis needs a series of studies to validate its efficiency and safety before recommending it as a therapeutic option.

In addition, a recent study found that vaginal samples with positive in vitro fertilization (IVF) clinical outcomes were significantly colonized by *Lactobacillus gasseri* and less colonized by *Bacteroides* and *Lactobacillus iners* [95]. This result may suggest the possibility of increasing the pregnancy rate by replenishing *Lactobacillus* in the vaginal cavity before IVF treatment by means of VMT.

Regulations and oversights, in FMT treatment, regarding donor selection, stool storage, recipient preparation, and delivery methods are becoming more sophisticated and generally accepted [71]. With more studies conducted, future VMT studies and clinical applications would be more regulated in donor selection, bacterial storage, and delivery methods.

## 8. Conclusions

After years of successful experience with FMT therapy, in recent years, VMT is becoming an available option for treatment for female reproductive issues, such as vaginosis. However, VMT is still in its infancy. Based on our understanding of the therapeutic potentiality of FMT, we can assume that the application of VMT may become a regular choice for treating and preventing many more female reproductive tract disorders. At the same time, due to the role of vaginal microbiota in a successful pregnancy, VMT may exert a potential role in increasing the reproductive rate. Therefore, VMT may become a popular choice of treatment in female patients with various issues.

The success of vaginal microbiota transplantation (VMT) is largely dependent on choosing an appropriate donor. An ideal vaginal microbiota composition is crucial in regulating disrupted microbiota in the recipient. Advancements in technology that enable the identification of suitable donors based on bacterial communities, combined with increasing research on the safety and efficacy of VMT, as well as established regulations and guidelines, can make the VMT process safer.

To make VMT more widely used in clinical therapy, it is important to have case-controlled studies as guidelines and regulatory approvals to ensure safety. Additionally, doctors and patients need clear and simple explanations of how VMT works. This paper provides information on the mechanisms involved in VMT treatment and future prospects for improving its clinical usage.

## Figures and Tables

**Figure 1 microorganisms-11-01427-f001:**
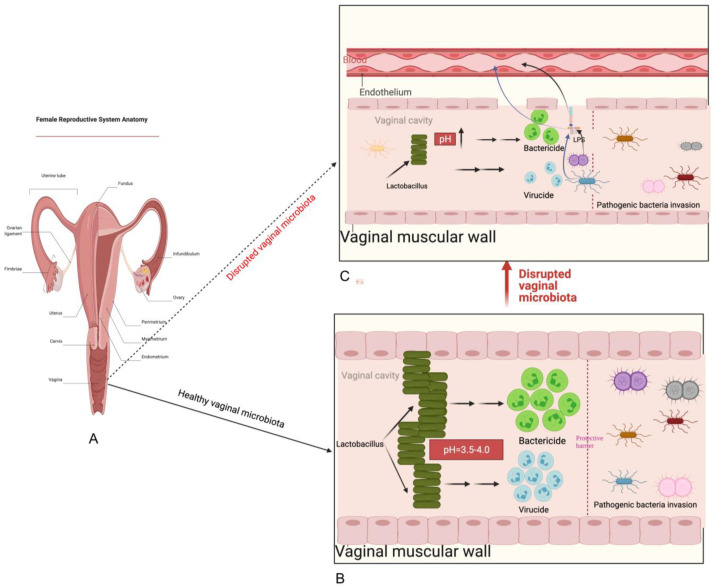
Overview of the female reproductive system. (**A**) female reproductive system anatomy. (**B**) illustration of a normal vaginal cavity microenvironment, which is characterized as rich in *Lactobacillus*, low in pH, enough amounts of bactericides and virucides, and a functional protective barrier. (**C**) illustration of a disrupted vaginal microenvironment, which is characterized by the lowered relative abundance of *Lactobacillus*, increased pH, lowered volume of bactericides and virucides, damaged protective barrier, and vaginal muscular wall. After these changes, the vagina becomes more susceptible to pathogenic agents’ invasion. Disruption of vaginal microbiota not only results in disease reactions in the vaginal cavity but also causes systemic inflammation when LPS produced by pathogenic bacteria enter the bloodstream via a damaged muscular wall and endothelium.

**Figure 2 microorganisms-11-01427-f002:**
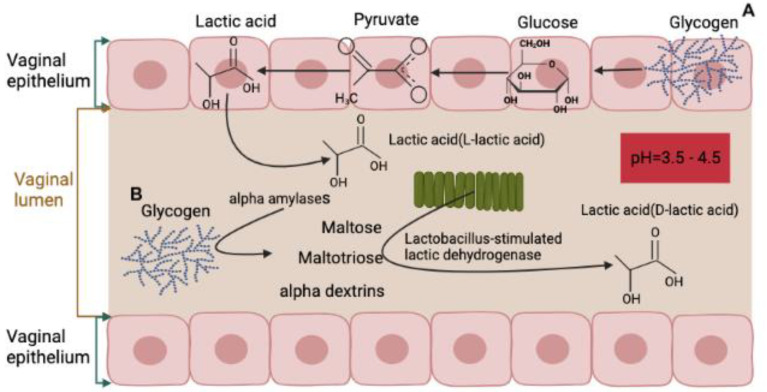
Vaginal lactic acid, key to vaginal homeostasis, is produced by two sources: the vaginal epithelium and the vaginal microbiota. A—The epithelium produces lactic acid through the conversion of glycogen to glucose, then to pyruvate, and finally to lactic acid, which is released into the vaginal lumen as the epithelium undergoes desquamation [50,51]. This process is controlled by estrogen levels in the blood [21] and is subject to change throughout a woman’s life cycle [50,52]. B—The second main source of lactic acid is from the conversion of glycogen found in the vaginal lumen by alpha-amylases to maltose, maltotriose, and alpha dextrins, then to lactic acid through the action of *Lactobacillus*-stimulated lactic dehydrogenase [50,52]. (Figure created using data from Barrientos-Durán, A. et al. [53]).

**Figure 3 microorganisms-11-01427-f003:**
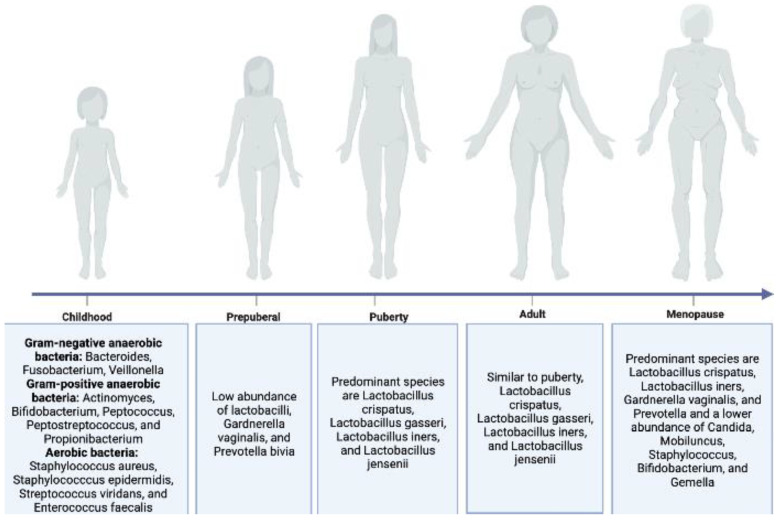
Vaginal microbiota composition throughout the female lifecycle, including childhood [54,55], prepuberty [55], puberty [56], adulthood [56], and menopause [57].

**Figure 4 microorganisms-11-01427-f004:**
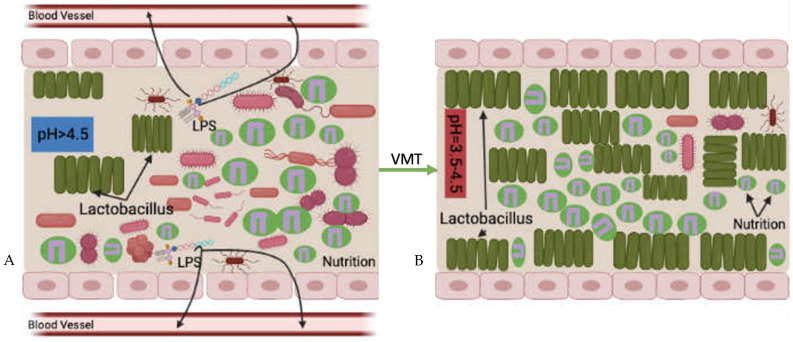
Potential mechanism of VMT; increased competition for nutrition ((**A**) before VMT; (**B**) after VMT).

**Figure 5 microorganisms-11-01427-f005:**
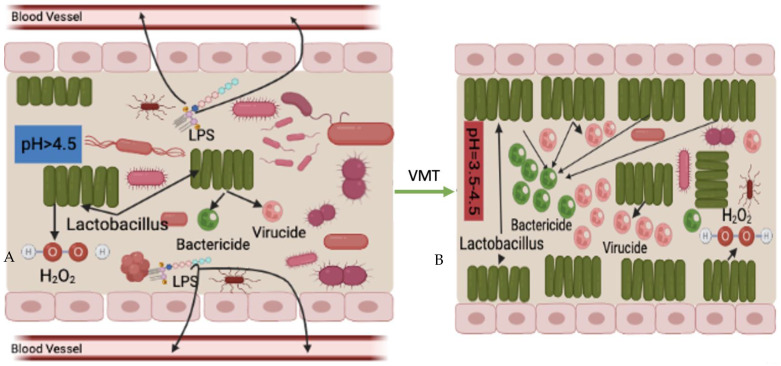
Potential mechanism of VMT; increased productions of bactericides and virucides, and H_2_O_2_. ((**A**) before VMT; (**B**) after VMT).

**Figure 6 microorganisms-11-01427-f006:**
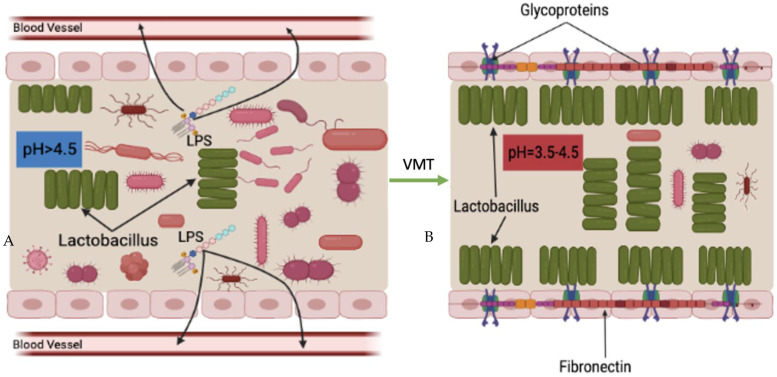
Potential mechanism of VMT; increased ability of epithelium adhesion. ((**A**) before VMT; (**B**) after VMT).

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
