# Peer review of "Possible Therapeutic Mechanisms and Future Perspectives of Vaginal Microbiota Transplantation"

_microorganisms, 2023, doi:10.3390/microorganisms11061427_

Round 1

Reviewer 1 Report

Tuniyazi et al. wrote a good review summarizing the potential therapeutic effect in VMT and three possible mechanisms behind. Although studies on VMT is still limited at this moment, the authors utilize the available resources to write a nice review to explore the mechanisms of VMT, the risks and limitations related to VMT, and the future development of VMT. This manuscript can be published after solving the following minor issues.

Minor issues:

1.     Line 44: Please provide the full name for the abbreviations “CDI”.

2.     Line 78-79: The phrase “predominantly composed of” is stated twice.

3.     Figure 1: Duplicated panel label: A, B, C. Label “C” blocked the words in the figure.

4.     Line 214-215 and Line 220-221: These two sentences are basically the same.

5.     Line 235 and line 244: The word “inhabit” should be “inhibit”.

6.     Line 349-350: Please provide the vaginal health status of the 31 female participants to demonstrate your statement “there are distinct microbial communities between healthy and women with reproductive problems.”

The English language is good.

Author Response

Reviewer 1:

  1. Line 44: Please provide the full name for the abbreviations “CDI”.

Dear reviewer, thank you for your time and comment. We corrected the text according to your suggestion. (Clostridium difficile infection(CDI) in 2012[13].)

  1. Line 78-79: The phrase “predominantly composed of” is stated twice.

Dear reviewer, thank you for your time and comment. We deleted “predominantly composed of” as you suggested.

  1. Figure 1: Duplicated panel label: A, B, C. Label “C” blocked the words in the figure.

Dear reviewer, thank you for your time and comment. We corrected the picture according to your suggestion.

  1. Line 214-215 and Line 220-221: These two sentences are basically the same.

Dear reviewer, thank you for your time and comment. We deleted Line 214-215 as you suggested.

  1. Line 235 and line 244: The word “inhabit” should be “inhibit”.

  1. Line 349-350: Please provide the vaginal health status of the 31 female participants to demonstrate your statement “there are distinct microbial communities between healthy and women with reproductive problems.”

Dear reviewer, thank you for your time and comment. We corrected the text according to your suggestion. (At the same time, studies also found that there are distinct microbial communities between healthy and women with reproductive problems. For example, a recent study involving 31 female participants revealed that they have different vaginal microbiota compositions[93]. This study showed that the vaginal microbiota was primarily characterized by the dominance of the genus Lactobacillus, specifically Lactobacillus iners AB-1, which was the most prevalent species among all the groups. Compared with the infertile cohort, the healthy group exhibited an excessive growth of anaerobic bacteria, including Leptotrichia and Snethia, which are associated with vaginal dysbiosis. )

Reviewer 2 Report

The authors have cited relevant literature to discuss the various aspects of Vaginal Microbiota Transfer (VMT) by discussing (a) the role of Lactobacilli in a healthy microbiota (b) the evolving microbiome during a woman's development and (c) risks and limitations of the therapy among others.

Some suggestions for the author include: 

1.  Figures need further editing, increased font size and resolution. 

2. Figure1: the figure appears to have duplicated A, B and C and disrupted microbiota appears cropped. If the figure has been modified from another paper, citations are needed accordingly. 

none

Author Response

  1. Figures need further editing, increased font size and resolution. 

Dear reviewer, thank you for your time and comment. We replaced Figure 1 to a better picture. Other pictures are already replaced with the highest resolution pictures.  

  1. Figure1: the figure appears to have duplicated A, B and C and disrupted microbiota appears cropped. If the figure has been modified from another paper, citations are needed accordingly. 

Dear reviewer, thank you for your time and comment. We corrected Figure 1 according to your suggestion. This picture is made by the author of the paper, no citation needed.
